# A critical residue in the $\alpha_1$M2–M3 linker regulating mammalian GABA$_A$ receptor pore gating by diazepam

Joseph W Nors, Shipra Gupta, Marcel P Goldschen-Ohm*

University of Texas at Austin, Department of Neuroscience, Austin, United States

**Abstract** Benzodiazepines (BZDs) are a class of widely prescribed psychotropic drugs that modulate activity of GABA$_A$ receptors (GABA$_A$Rs), neurotransmitter-gated ion channels critical for synaptic transmission. However, the physical basis of this modulation is poorly understood. We explore the role of an important gating domain, the $\alpha_1$M2–M3 linker, in linkage between the BZD site and pore gate. To probe energetics of this coupling without complication from bound agonist, we use a gain of function mutant ($\alpha_1$L9'T$\beta_2\gamma_{2L}$) directly activated by BZDs. We identify a specific residue whose mutation ($\alpha_1$V279A) more than doubles the energetic contribution of the BZD positive modulator diazepam (DZ) to pore opening and also enhances DZ potentiation of GABA-evoked currents in a wild-type background. In contrast, other linker mutations have little effect on DZ efficiency, but generally impair unliganded pore opening. Our observations reveal an important residue regulating BZD-pore linkage, thereby shedding new light on the molecular mechanism of these drugs.

## Introduction

Benzodiazepines (BZDs) (e.g. Valium, Xanax) are one of the most widely prescribed psychotropic drugs today. An estimated nearly 100 million adults in the United States are prescribed a BZD annually (*Agarwal and Landon, 2019*; *Bachhuber et al., 2016*; *Olfson et al., 2015*). Their anxiolytic and sedative properties are used as therapies for conditions including anxiety, panic, insomnia, seizures, muscle spasms, pain, and alcohol withdrawal (*Möhler et al., 2002*). Although largely effective, BZDs have undesirable effects, including tolerance, addiction, dependence, and withdrawal symptoms, and are often co-abused with alcohol and opioids (*Fluyau et al., 2018*; *Schmitz, 2016*; *Jones and McAninch, 2015*; *Jones et al., 2010*). Novel therapies with reduced risks are imperative for safer long-term treatment options.

The therapeutic effects of BZDs are conferred upon binding to and modulating the activity of GABA$_A$Rs, which are the primary inhibitory neurotransmitter-gated ion channels in the central nervous system (*Smart and Stephenson, 2019*). GABA$_A$Rs are part of the Cys-loop superfamily of pentameric ligand-gated ion channels (pLGICs) including homologous glycine (Gly), nicotinic acetylcholine (nACh), serotonin type 3 (5-HT$_3$), and zinc-activated receptors as well as prokaryotic homologs (*Pless and Sivilotti, 2018*; *Nemecz et al., 2016*; *Tasneem et al., 2004*; *Connolly and Wafford, 2004*). Each pentameric GABA$_A$R is comprised of subtype-specific combinations of five homologous but nonidentical subunits ($\alpha_{1-6}$, $\beta_{1-3}$, $\gamma_{1-3}$, $\delta$, $\varepsilon$, $\pi$, $\theta$, $\rho_{1-3}$) that together form a central chloride conducting pore (*Figure 1*; *Olsen and Sieghart, 2008*). The most prominent subtype at synapses is comprised of $\alpha_1$, $\beta_n$, and $\gamma_2$ subunits (*Sieghart and Sperk, 2002*). GABA$_A$Rs provide a critical balance with excitatory signaling for normal neural function, and not surprisingly, their dysfunction is related to disorders such as epilepsy, autism spectrum disorder, intellectual disability, schizophrenia, and neurodevelopmental disorders such as fragile X syndrome (*Hernandez and Macdonald, 2019*; *Gao et al., 2018*; *Mahdavi et al., 2018*; *Braat and Kooy, 2015*; *Vien et al., 2015*;

*For correspondence:
marcel.goldschen-ohm@austin.utexas.edu

Competing interests: The authors declare that no competing interests exist.

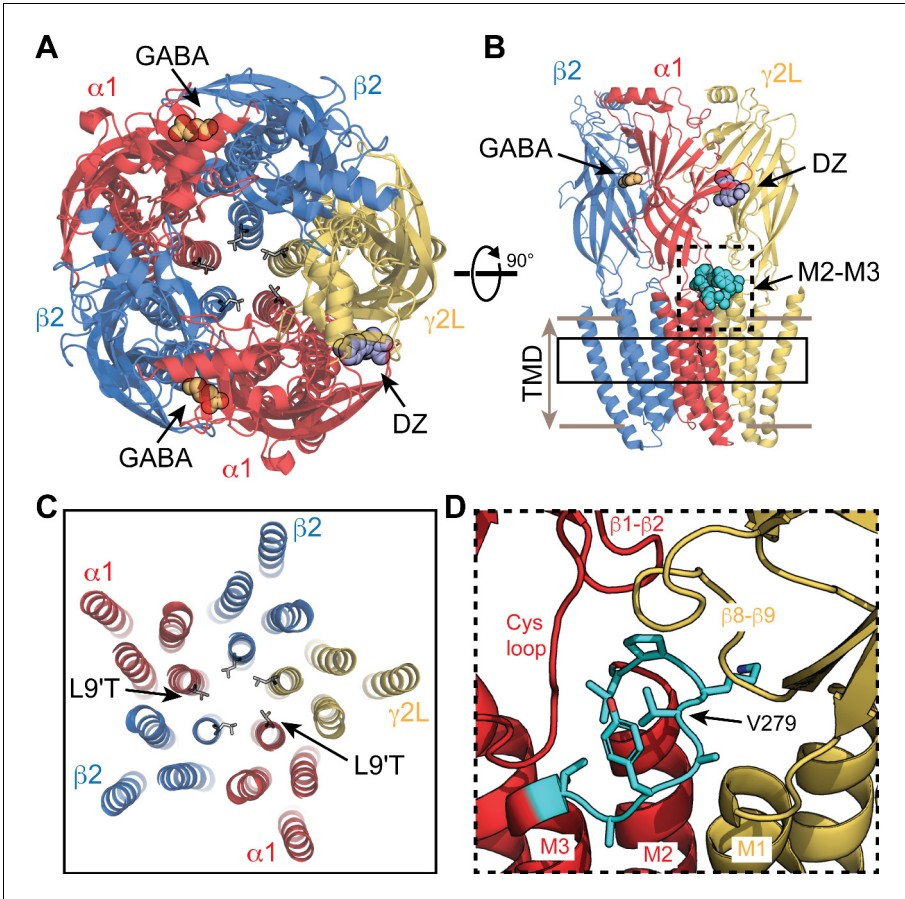

**Figure 1.** Visual representation of an $\alpha_1\beta_2\gamma_2$ GABA$_A$ receptor from cryo-EM map PDB 6X3X. (A,B) View from the extracellular space (A) and parallel to the membrane plane (B). Bound GABA and DZ are shown as gold and lavender spheres, respectively. The 9' pore residue from each subunit is shown in stick representation, all leucine except for the mutation $\alpha_1$L9'T, which was generated in PyMol as a visualization aid. One of the two $\alpha_1$M2–M3 linkers is shown as cyan spheres in (B). (C) Same view as in (A) for a slice through the transmembrane domains indicated by the solid box in (B). (D) Detail for the dashed box in (B). The $\alpha_1$M2–M3 linker (L276-T283; rat numbering) is colored cyan with side chains shown in stick representation.

The online version of this article includes the following figure supplement(s) for figure 1:

**Figure supplement 1.** Sequence alignment of M2–M3 linker regions for subunits from several members of the pLGIC superfamily.

*Rudolph and Möhler, 2014*; *Limon et al., 2012*; *Ramamoorthi and Lin, 2011*; *Solís-Añez et al., 2007*). Although pharmacological manipulation of GABA$_A$Rs is a powerful approach to tuning neural signaling, the rational design of novel therapies is challenged by a lack of understanding of the molecular mechanism by which drugs such as BZDs modulate channel behavior.

In conjunction with structural models of mostly homomeric pLGICs, recent cryo-EM models of heteromeric GABA$_A$Rs with bound neurotransmitter and BZD provide an important conceptual aid to understand the mechanism of action of these drugs (*Kim et al., 2020*; *Masiulis et al., 2019*; *Laverty et al., 2019*; *Phulera et al., 2018*; *Zhu et al., 2018*). Consistent with earlier functional studies (*Sigel and Ernst, 2018*; *Sigel and Lüscher, 2011*; *Sigel, 2002*; *Wagner and Czajkowski, 2001*; *Kucken et al., 2000*; *McKernan et al., 1998*; *Pritchett et al., 1989*; *Gibbs et al., 1985*), they show that BZDs are bound at a specific recognition site in the extracellular domain between $\alpha$ and $\gamma$ subunits (*Figure 1*). However, these structures have yet to clarify the mechanism by which BZDs modulate channel activity.

Kinetic models of interacting domains and thermodynamic linkage analysis can be used to estimate the energy of interaction between binding sites and the pore gate from bulk measures of

ensemble average activity (*Goldschen-Ohm et al., 2014*; *Chowdhury and Chanda, 2013*; *Wyman and Gill, 1990*). Coupled with mutagenesis or other perturbations, these interactions can be probed to elucidate their physical basis. However, these approaches are challenged for BZDs because they do not evoke robust channel opening by themselves, but instead modulate responses to an agonist such as GABA. Thus, typical experimental measures reflect channels with both agonist and BZD bound. This makes it difficult to dissect whether the drug has altered either agonist binding or channel gating as the two processes are intimately coupled (*Colquhoun, 1998*). This challenge has contributed to differing conclusions postulating that BZDs alter either agonist binding (*Goldschen-Ohm et al., 2010*; *Perrais and Ropert, 1999*; *Mellor and Randall, 1997*; *Rogers et al., 1994*; *Vicini et al., 1987*), pore gating (*Li et al., 2013*; *Downing et al., 2005*; *Campo-Soria et al., 2006*; *Rüsch and Forman, 2005*), or an intermediate preactivation step (*Goldschen-Ohm et al., 2014*; *Gielen et al., 2012*). The ability of BZDs to potentiate current responses to saturating concentrations of partial agonists, and also to directly gate gain of function mutants, implies that the drug's effect is not due to changes in binding alone. A combination of effects on both binding and gating as predicted by changes in intermediate gating steps energetically coupled with both binding sites and the pore gate is plausible (*Goldschen-Ohm et al., 2014*). However, the molecular identity of any such intermediate states remains unclear.

Here we examined linkage between the BZD site and pore gate in isolation from any effect of the drug on agonist binding using a spontaneously active gain of function mutant ($\alpha_1$L9'T$\beta_2\gamma_{2L}$) that is directly gated by BZDs alone (*Scheller and Forman, 2002*; *Chang and Weiss, 1999*). In the $\alpha_1$L9'T background, we serially mutated each residue in the $\alpha_1$M2–M3 linker to alanine and assessed the effect on modulation of the channel pore by the BZD positive modulator DZ (Valium) in the absence of agonist. The M2–M3 linker is a loop following the pore lining M2 helix that together with several other important interfacial loops defines the region connecting extracellular ligand binding and transmembrane domains known to be crucially involved in the gating process of pLGICs (*Figure 1*). Structural models show that agonist binding is associated with an outward displacement of the M2–M3 linker from the central pore axis (*Nemecz et al., 2016*). Mutations in the linker generally impair gating by agonists (*Kash et al., 2003*; *O'Shea and Harrison, 2000*), and some are associated with genetic diseases such as epilepsy in GABA$_A$Rs (*Hales et al., 2006*; *Hernandez and Macdonald, 2019*) or hyperekplexia in GlyRs (*Bode and Lynch, 2014*). However, the role of the M2–M3 linker in drug modulation by BZDs is less understood.

We show that alanine substitutions throughout the $\alpha_1$M2–M3 linker generally impair unliganded pore gating, whereas they have little effect on the efficiency by which chemical energy from DZ binding is transmitted to the pore gate. The notable exception is $\alpha_1$V279A, which more than doubles DZ's energetic contribution to pore opening, whereas larger side chains at this site do not. In a wild-type background, $\alpha_1$V279A enhances DZ potentiation of currents evoked by even saturating GABA, consistent with a direct effect on the pore closed/open equilibrium. Our observations identify an important residue in the $\alpha_1$M2–M3 linker regulating the efficiency of BZD modulation in GABA$_A$Rs.

## Results

### Alanine substitutions in the $\alpha_1$M2–M3 linker generally impair unliganded gating

We use the gain of function mutation $\alpha_1$L9'T that confers spontaneous channel gating in the absence of agonist that can be further modulated by BZDs (*Scheller and Forman, 2002*; *Chang and Weiss, 1999*). The primary purpose for this mutation is to enable macroscopic current responses to report on pore gating by a BZD in the absence of agonist. Initially, we examine unliganded and GABA-evoked gating and in later sections turn to BZDs. In the $\alpha_1$L9'T$\beta_2\gamma_{2L}$ ($\alpha_1$L9'T or L9'T) gain of function background, we assessed the effects of alanine substitutions in the $\alpha_1$M2–M3 linker on unliganded and GABA-evoked channel activity. We restricted the scan primarily to the flexible loop between helical regions associated with M2 and M3 from $\alpha_1$L276-T283 (*Figure 1*; *Figure 1—figure supplement 1*). Two of these positions, $\alpha_1$A280 and $\alpha_1$A282, are natively alanine and were not mutated. Briefly, *Xenopus laevis* oocytes were co-injected with mRNA for $\alpha$, $\beta$, and $\gamma$ subunits in a 1:1:10 ratio, and current responses to microfluidic application of ligands were recorded in two-electrode voltage clamp. Each recording consisted of a series of 10 second pulses of GABA at various concentrations

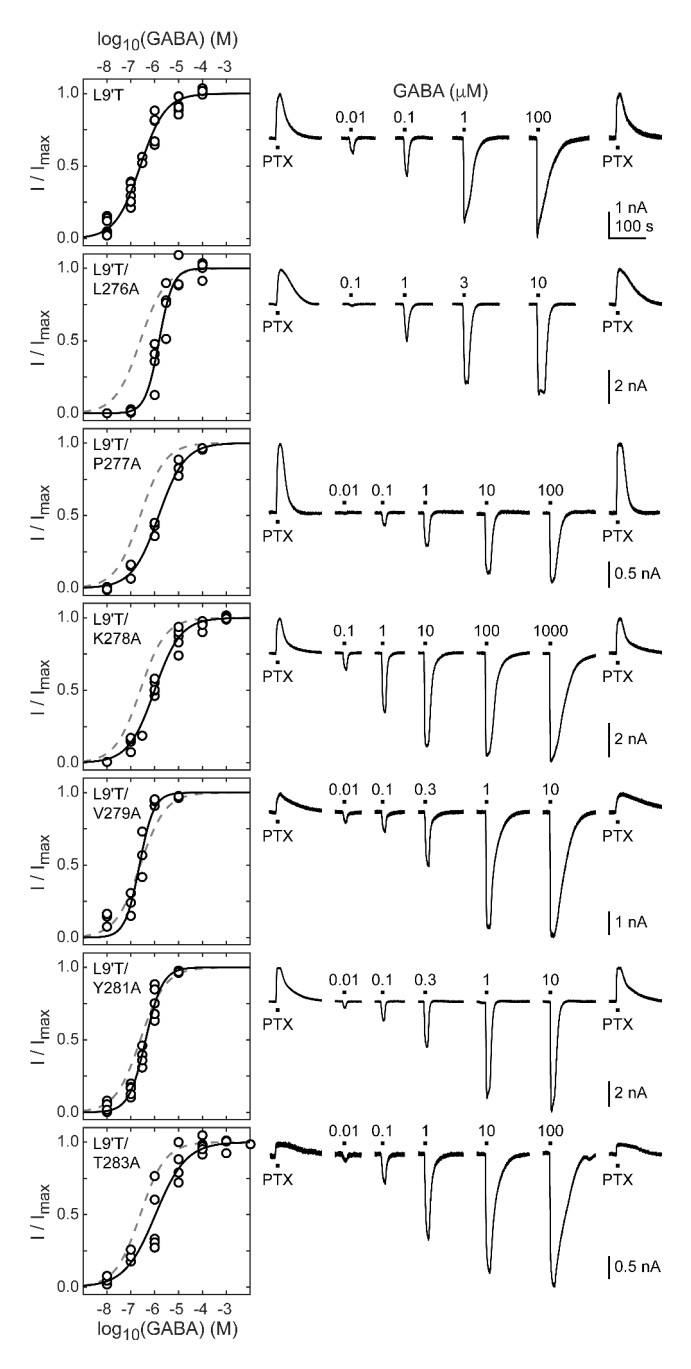

**Figure 2.** Spontaneous PTX-sensitive and GABA-evoked currents for $\alpha_1$M2–M3 linker alanine substitutions in the gain of function $\alpha_1$L9'T$\beta_2\gamma_{2L}$ background. (Left) Summary of normalized GABA concentration–response relationships for GABA-evoked currents with the zero current baseline set to the level of spontaneous activity. Solid line is a fit of the pooled data across oocytes to *Equation 1*, and the dashed line is the fit for the L9'T background. Fit parameters are EC$_{50}$, hill slope (# oocytes): L9'T = 0.25 µM, 0.83 (7); L9'T/L276A = 1.53 µM, 1.51 (4); L9'T/P277A = 1.48 µM, 0.80 (3); L9'T/K278A = 0.99 µM, 0.79 (5); L9'T/V279A = 0.23 µM, 1.44 (4); L9'T/Y281A = 0.42 µM, 1.23 (5); L9'T/T283A = 1.08 µM, 0.67 (5). Parameters for fits to individual oocytes are summarized in *Figure 2—figure supplement 2*. (Right) Example currents in response to 10 second pulses of the pore blocker PTX (1 mM) or GABA (concentration in micromolar indicated above each pulse). Responses to GABA were bookended by application of PTX to assess the amount of spontaneous current and to normalize any drift or rundown during the experiment (see *Figure 2—figure supplement 1*).

The online version of this article includes the following figure supplement(s) for figure 2:

*Figure 2 continued on next page*

*Figure 2 continued*

**Figure supplement 1.** Correction for baseline drift and current rundown.
**Figure supplement 2.** Summary of GABA $EC_{50}$ and hill slope from fitting *Equation 1* to GABA concentration–response relationships as shown in *Figure 2* for individual oocytes.
**Figure supplement 3.** Responses of wild-type $\alpha_1\beta_2\gamma_{2L}$ receptors to 10 second pulses of either 1 mM PTX, 3 μM DZ, or 3 mM GABA.

bookended by 10 second pulses of 1 mM picrotoxin (PTX) (*Figure 2*). Current block by the pore blocker PTX was used to assess the amount of spontaneous activity and to correct for drift or rundown over the course of the experiment (see *Figure 2—figure supplement 1*). In comparison, wild-type receptors have little to no PTX-sensitive spontaneous activity (*Figure 2—figure supplement 3*). Normalized concentration–response relationships for the additional GABA-evoked current above the spontaneous current baseline in the L9'T background are shown in *Figure 2*. As compared to $\alpha_1$L9'T, all of the alanine mutations exhibited an increase in the $EC_{50}$ and/or steepness of the GABA concentration–response relationship (*Figure 2*; *Figure 2—figure supplement 2*). The increased $EC_{50}$ is generally consistent with previous reports of $\alpha_1$M2–M3 linker mutations in a wild-type background (*O'Shea and Harrison, 2000*) and implies that side chain interactions in this region are important for channel gating by agonists. The steepness of the concentration–response relationship for $\alpha_1$L9'T was fairly shallow, with a hill slope slightly below one. This could reflect an increased efficiency of mono-liganded gating in the gain of function background. In such a case, the steeper hill slopes conferred by $\alpha_1$L276A, $\alpha_1$V279A, and $\alpha_1$Y281A may be due to a reduction in the efficiency of mono- vs di-liganded gating, although this was not verified.

The level of unliganded channel activity was determined from the ratio of the spontaneous current amplitude as assessed by block with PTX ($I_{PTX}$) to the maximal current amplitude evoked with saturating GABA ($I_{GABA-max}$) (*Figure 3A*). The unliganded open probability was estimated as the product of the ratio $I_{PTX}/I_{GABA-max}$ and the open probability in saturating GABA ($P_{o-GABA-max}$). Under the assumption that $P_{o-GABA-max}$ is similar for all constructs, all $\alpha_1$M2–M3 linker mutations except for $\alpha_1$P277A reduced the unliganded open probability by approximately two-fold (*Figure 3B*, *Table 1*). $P_{o-GABA-max}$ is ~0.85 in wild-type channels (*Keramidas and Harrison, 2010*), and likely to be closer to 1.0 in a gain of function background such as $\alpha_1$L9'T that is further known to desensitize much more slowly than wild-type (*Scheller and Forman, 2002*). This assumption was verified for single

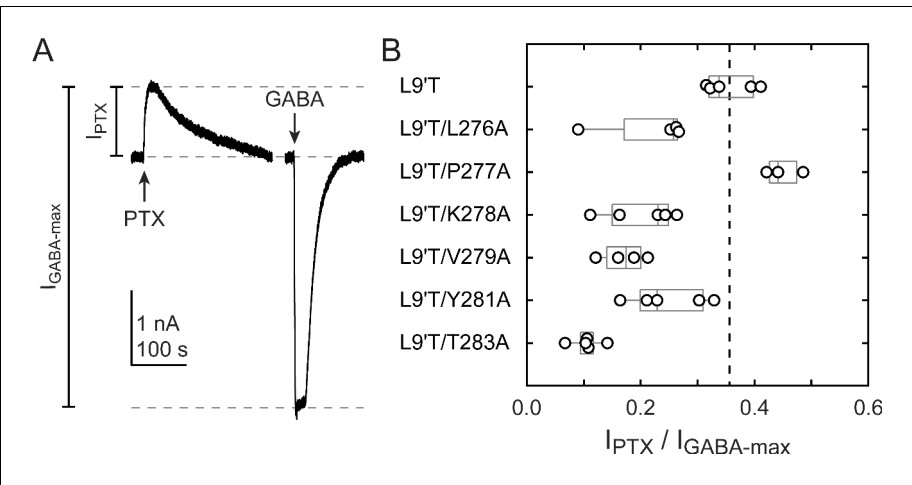

**Figure 3.** Ratio of PTX-sensitive to maximal GABA-evoked current amplitude. (**A**) Example currents from $\alpha_1$L9'T/V279A$\beta_2\gamma_{2L}$ receptors elicited by 10 second pulses of either 1 mM PTX or 10 μM GABA. (**B**) Summary of the ratio of PTX-sensitive to maximal GABA-evoked current amplitude for individual oocytes. Gray box plots indicate the median and 25th and 75th percentiles. The vertical dashed line is the mean for L9'T. These ratios were estimated as being approximately proportional to the unliganded open probability by a factor that is the open probability in saturating GABA.

**Table 1.** Summary of peak current ratios and $\Delta\Delta G_{DZ}$ for mutations in the L9'T background. Data are mean ± standard deviation (# oocytes), and individual data points are shown in *Figures 3B*, *5B*, *6C,* and *7C*.

| | $I_{PTX}$ /$I_{GABA-max}$ | $I_{DZ-max}$ /$I_{PTX}$ | $\Delta\Delta G_{DZ}$ (kcal/mol) |
|---|---|---|---|
| L9'T | 0.36 ± 0.04 (5) | 1.35 ± 0.05 (5) | −0.31 ± 0.04 (5) |
| L9'T/L276A | 0.22 ± 0.09 (4) | 1.62 ± 0.07 (5) | −0.45 ± 0.04 (5) |
| L9'T/P277A | 0.45 ± 0.03 (3) | 1.41 ± 0.17 (5) | −0.46 ± 0.20 (5) |
| L9'T/K278A | 0.20 ± 0.06 (5) | 1.67 ± 0.15 (4) | −0.44 ± 0.08 (4) |
| L9'T/V279A | 0.17 ± 0.04 (4) | 3.30 ± 0.70 (5) | −1.11 ± 0.30 (5) |
| L9'T/V279D | 0.02 ± 0.005 (3) | 2.31 ± 0.08 (3) | −0.52 ± 0.02 (3) |
| L9'T/V279W | 0.16 ± 0.05 (2) | 1.96 ± 0.23 (3) | −0.53 ± 0.10 (3) |
| L9'T/Y281A | 0.25 ± 0.07 (5) | 1.59 ± 0.11 (4) | −0.42 ± 0.07 (4) |
| L9'T/T283A | 0.11 ± 0.03 (5) | 1.71 ± 0.10 (6) | −0.38 ± 0.04 (6) |

L9'T/V279A receptors (*Figure 7—figure supplement 1*). Thus, any reduction in $P_{o\text{-}GABA\text{-}max}$ for the linker mutations should only further reduce their estimated unliganded open probability. In summary, these data show that alanine mutations in the $\alpha_1$M2–M3 linker generally impair channel gating by increasing GABA $EC_{50}$ and/or reducing unliganded pore opening. This suggests that $\alpha_1$M2–M3 linker side chain interactions play an important role in the closed vs. open pore equilibrium even in the absence of agonist.

## All but one alanine substitution in the $\alpha_1$M2–M3 linker have little effect on activation by DZ relative to unliganded activity

BZD positive modulators alone evoke channel opening with extremely low probability (*Campo-Soria et al., 2006*), thus necessitating coapplication with an agonist to obtain robust currents from wild-type receptors (*Germann et al., 2019*). However, dissecting the effects of BZDs on either agonist binding or channel gating is severely challenged in the presence of an agonist because the two processes are intimately coupled (*Colquhoun, 1998*). To isolate effects on gating apart from any effects on agonist binding, we examined the effect of alanine substitutions in the $\alpha_1$M2–M3 linker on DZ-evoked currents in the background of the gain of function mutation $\alpha_1$L9'T in the absence of agonist. Concentration–response relationships for currents evoked by 10 second pulses of DZ were bookended by 10 second pulses of 1 mM PTX to assess spontaneous activity (*Figure 4*). Drift or run-down was corrected as described for GABA-evoked currents (*Figure 2—figure supplement 1*). Normalized concentration–response relationships for the additional DZ-elicited current above the spontaneous current baseline were fit to *Equation 1*. Alanine substitutions in the $\alpha_1$M2–M3 linker have no obvious effect on DZ concentration–response relationships with the exception of $\alpha_1$L276A that confers a right shift and reduction in steepness (*Figure 4*; *Figure 4—figure supplement 1*). Reports for the $EC_{50}$ of DZ potentiation in wild-type receptors are similar to the DZ $EC_{50}$ for $\alpha_1$L9'T receptors reported here and in previous studies (*Li et al., 2013*; *Campo-Soria et al., 2006*; *Rüsch and Forman, 2005*; *Walters et al., 2000*).

The ratio of the maximal DZ-evoked current amplitude ($I_{DZ-max}$) to PTX-sensitive current amplitude ($I_{PTX}$) is a measure for how well DZ activates the channel relative to its spontaneous unliganded activity. This ratio is largely independent of alanine substitution in the $\alpha_1$M2–M3 linker with the notable exception of $\alpha_1$V279A (*Figure 5*, Table 1). Thus, apart from $\alpha_1$V279A, DZ-evoked currents are predictably proportional to the amount of unliganded activity. This suggests that $\alpha_1$V279 has a unique role in DZ modulation, whereas side chains of other linker residues are involved little or not at all.

## Dependence of DZ gating on charge and volume at $\alpha_1$V279

The M2–M3 linker has both high sequence and structural similarity in pLGICs. The valine at position 279 in the $GABA_A$R $\alpha_1$ subunit is located near the center of the linker and is nearly always an aliphatic residue in different $GABA_A$R subunits as well as in subunits of other pLGICs, with the exception of AChR, where it is a threonine. Even where sequences differ, structural models of the M2–M3

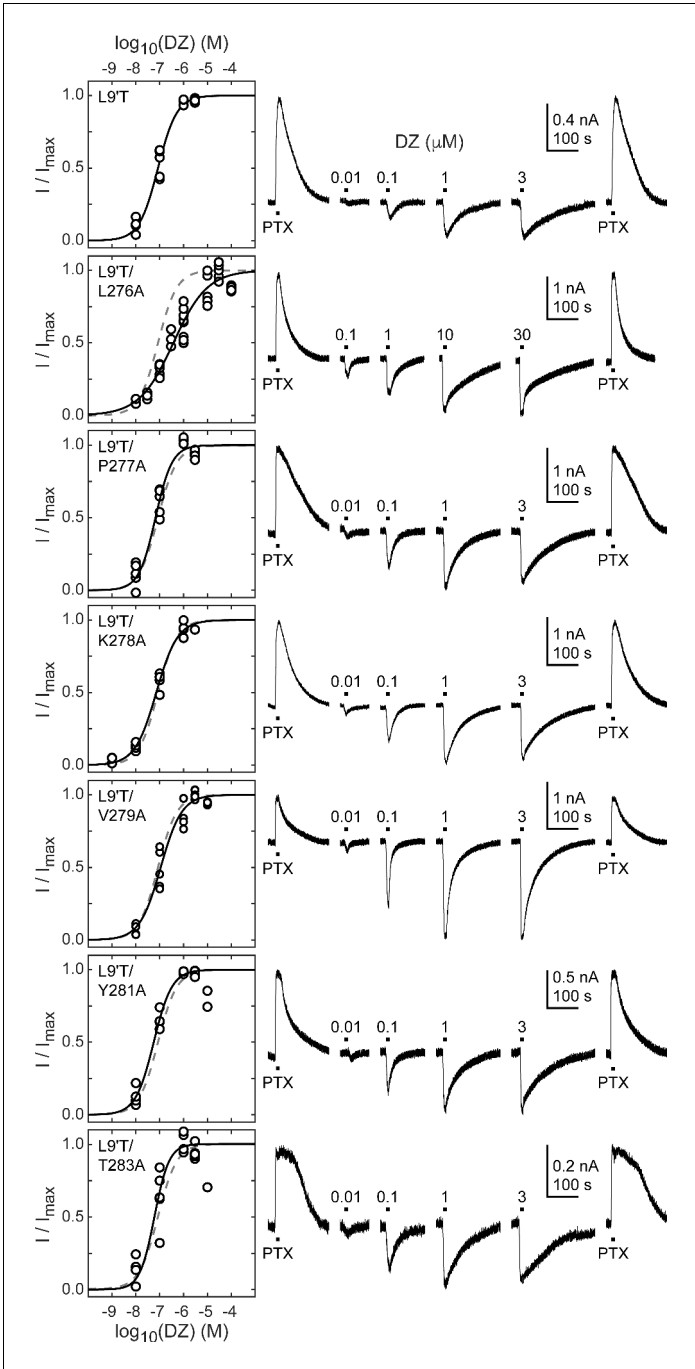

**Figure 4.** Spontaneous PTX-sensitive and DZ-evoked currents for $\alpha_1$M2–M3 linker alanine substitutions in the gain of function $\alpha_1$L9'T$\beta_2\gamma_{2L}$ background. (Left) Summary of normalized DZ concentration–response relationships for DZ-evoked currents with the zero current baseline set to the level of spontaneous activity. Solid line is a fit of the pooled data across oocytes to *Equation 1*, and the dashed line is the fit for the L9'T background. Reduced peak responses to 10 µM DZ for L9'T/Y281A and L9'T/T283A were omitted from the fits. Fit parameters are $EC_{50}$, hill slope (# oocytes): L9'T = 85 nM, 1.10 (5); L9'T/L276A = 380 nM, 0.61 (8); L9'T/P277A = 67 nM, 1.24 (5); L9'T/K278A = 72 nM, 0.97 (4); L9'T/V279A = 110 nM, 0.98 (5); L9'T/Y281A = 56 nM, 1.14 (4); L9'T/T283A = 60 nM, 1.42 (6). Parameters for fits to individual oocytes are summarized in *Figure 4—figure supplement 1*. (Right) Example currents in response to 10 second pulses of the pore blocker PTX (1 mM) or DZ (concentration in micromolar indicated above each pulse). Responses to DZ were bookended by application of PTX to assess the amount of spontaneous current and to normalize any drift or rundown during the experiment (see *Figure 2—figure supplement 1*).

*Figure 4 continued on next page*

*Figure 4 continued*

The online version of this article includes the following figure supplement(s) for figure 4:

**Figure supplement 1.** Summary of DZ EC$_{50}$ and hill slope from fitting *Equation 1* to DZ concentration–response relationships as shown in *Figure 4* for individual oocytes.

linker for multiple pLGICs are highly similar. To explore the side chain properties relevant to DZ gating, we examined the effect of introducing a charged aspartate or bulky tryptophan at position 279 in the $\alpha_1$L9'T background. We recorded unliganded PTX-sensitive and GABA- or DZ-evoked currents as described above and compared their relative current amplitudes and concentration–response relationships (*Figure 6*). The negative charge introduced by $\alpha_1$V279D severely inhibits unliganded gating, exhibiting very little PTX-sensitive current in relation to robust currents evoked with saturating GABA. In contrast, addition of the bulky side chain $\alpha_1$V279W only slightly impairs unliganded gating to a similar degree as substitution with alanine. These data suggest that side chain volume at this position is less critical for unliganded gating, whereas introduction of a negative charge nearly abolishes spontaneous activity despite the $\alpha_1$L9'T pore mutation. Consistent with impaired pore gating, $\alpha_1$V279D increases the GABA EC$_{50}$ by ~100-fold.

The ratio of DZ-evoked to PTX-sensitive current amplitude is only slightly increased by $\alpha_1$V279W, whereas it is further increased by $\alpha_1$V279D, albeit not to the extent of $\alpha_1$V279A (*Figure 6C*). However, we were unable to reach saturation for DZ-evoked currents from $\alpha_1$V279D due to a reduction in peak current amplitude at higher DZ concentrations consistent with occupation of a secondary lower affinity inhibitory site (*Figure 6B*). Thus, our observations for $\alpha_1$V279D reflect a lower limit on channel activity evoked by DZ binding. Even if $\alpha_1$V279D has an appreciable effect on the ratio of DZ-evoked to unliganded current, the overall effect of this mutation on channel function is likely to be dominated by its inhibition of pore gating. In contrast, $\alpha_1$V279W has similar effects to alanine substitutions at other linker residues. These data suggest that DZ gating is specifically enhanced by a reduction of side chain volume at position 279.

## The mutation $\alpha_1$V279A increases DZ's energetic contribution to pore gating

To estimate the amount of chemical energy from DZ binding that is transmitted to the pore gate, we employed a simple model of channel gating between closed (C) and open (O) pore states in both unliganded and DZ-bound conditions (*Figure 7A*). The model assumes that gating can be

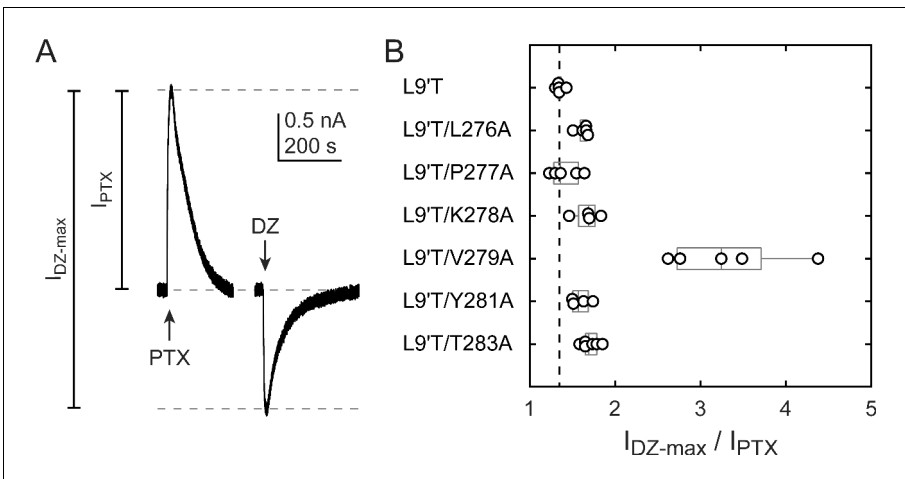

**Figure 5.** Ratio of maximal DZ-evoked to PTX-sensitive current amplitude. (**A**) Example currents from $\alpha_1$L9'T/K278A$\beta_2\gamma_{2L}$ receptors elicited by 10 second pulses of either 1 mM PTX or 1 µM DZ. (**B**) Summary of the ratio of maximal DZ-evoked to PTX-sensitive current amplitude for individual oocytes. Gray box plots indicate the median and 25th and 75th percentiles. The vertical dashed line is the mean for L9'T. These ratios were used as estimates for the approximate fold-change in open probability upon DZ binding.

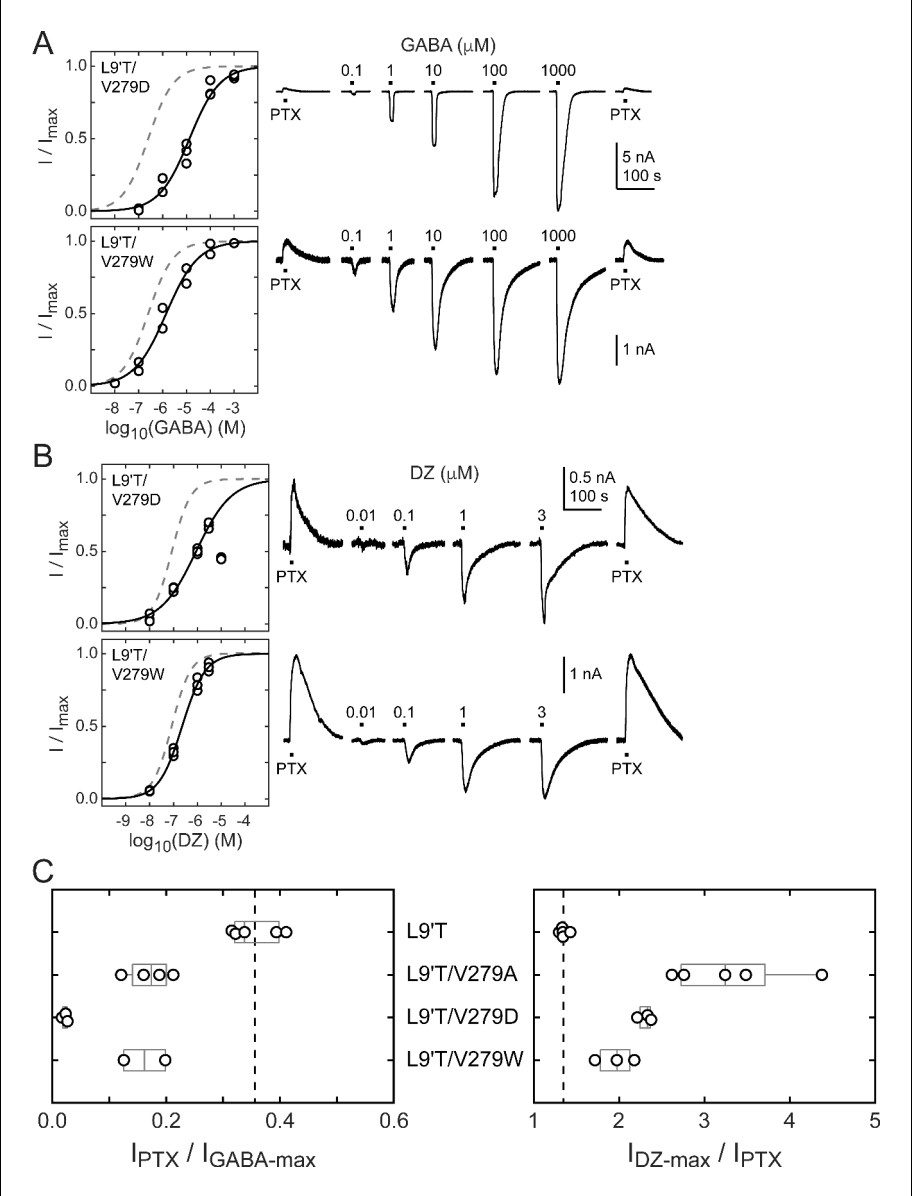

**Figure 6.** Spontaneous PTX-sensitive and GABA- or DZ-evoked currents for $\alpha_1$V279D and $\alpha_1$V279W in the gain of function $\alpha_1$L9'T$\beta_2\gamma_{2L}$ background. (**A**) Summary of normalized GABA concentration–response relationships for GABA-evoked currents with the zero current baseline set to the level of spontaneous activity (left). Solid line is a fit of the combined data across oocytes to *Equation 1*, and the dashed line is the fit for the L9'T background. Fit parameters are $EC_{50}$, hill slope (# oocytes): L9'T/V279D = 13 μM, 0.68 (3); L9'T/V279W = 1.4 μM, 0.65 (2). Example currents in response to 10 second pulses of the pore blocker PTX (1 mM) or GABA (concentration in micromolar indicated above each pulse) (right). (**B**) Same as in (**A**) except for DZ-evoked currents. Responses from L9'T/V279D to 10 μM DZ were excluded from the fit. Fit parameters are $EC_{50}$, hill slope (# oocytes): L9'T/V279D = 0.88 μM, 0.58 (3); L9'T/V279W = 0.23 μM, 0.90 (3). (**C**) Summary of the ratios of PTX-sensitive and either maximal GABA- or DZ-evoked current amplitudes for individual oocytes. Gray box plots indicate the median and 25th and 75th percentiles. The vertical dashed line is the mean for L9'T. These ratios were used as estimates for the approximate fold-change in open probability upon DZ binding.

approximated with a single closed and open states in each liganded condition. The free energy differences for pore gating in unliganded ($\Delta G_0$) and DZ-bound ($\Delta G_1$) conditions were calculated according to *Equations 2–3* under the assumption that $P_{o\text{-GABA-max}} \sim 1$. A comparison of $\Delta G_0$ versus $\Delta G_1$ shows that DZ binding confers a uniform $\Delta\Delta G_{DZ} = \Delta G_1 - \Delta G_0 = -0.4$ kcal/mol to the pore gating

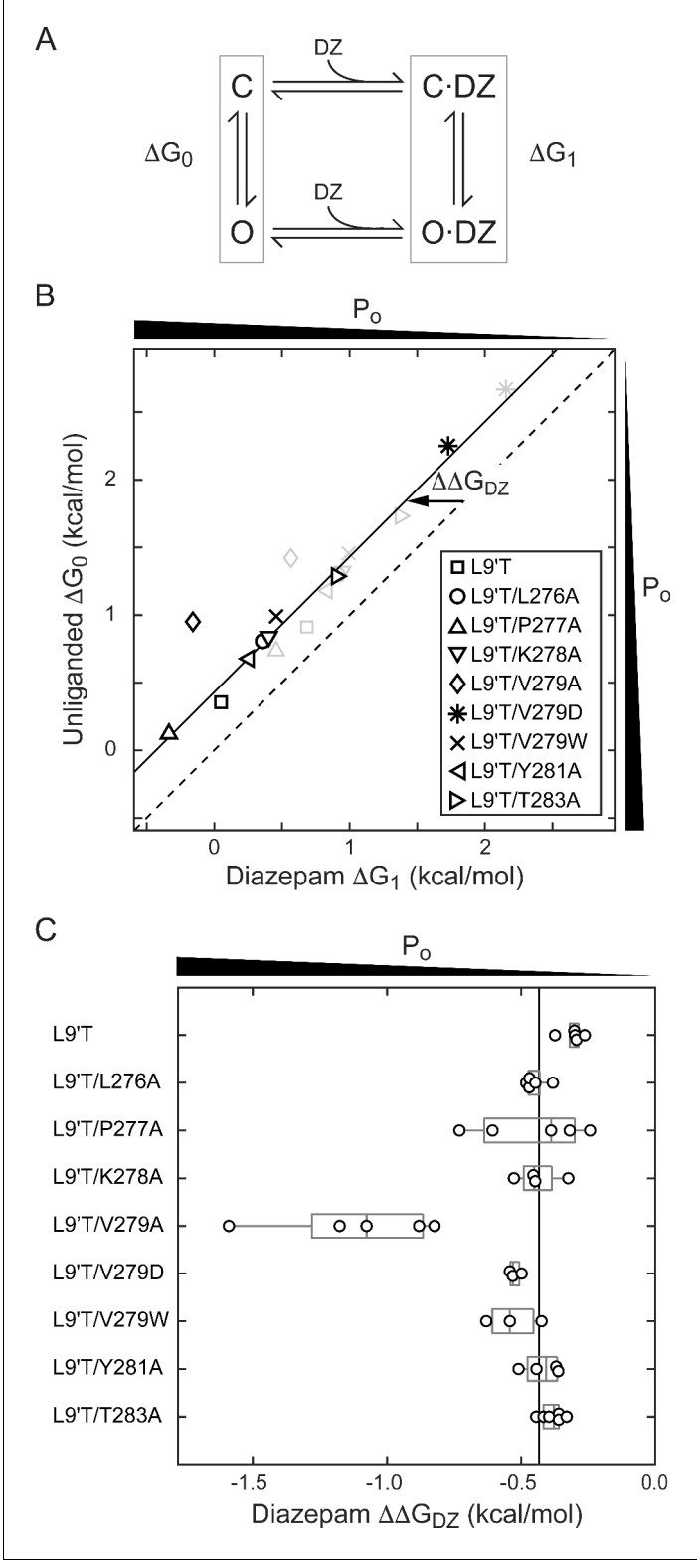

**Figure 7.** A critical residue in the $\alpha_1$M2–M3 linker regulating DZ's energetic contribution to pore gating. (**A**) A simple model approximating channel gating between closed (C) and open (O) pore states in both unliganded and DZ-bound conditions. (**B**) Relationship between gating free energy in unliganded ($\Delta G_0$) and DZ-bound ($\Delta G_1$) conditions for $\alpha_1$L9'T$\beta_2\gamma_{2L}$ receptors and $\alpha_1$M2–M3 linker mutations assuming $P_{o–GABA–max} = 1$ (bold symbols) or $P_{o–GABA–max} = 0.5$ (light gray symbols) (Equations 2–3). $\Delta G$ values are the change in energy from closed to open

*Figure 7 continued on next page*

*Figure 7 continued*

states, such that negative values favor opening. To illustrate this, we indicate the direction of increasing open probability ($P_o$) along each axis. Points are the mean across oocytes. The dashed line of symmetry reflects $\Delta G_0 = \Delta G_1$ where DZ would have no effect on pore gating. The solid line is a fit to $\Delta G_1 = \Delta G_0 + \Delta\Delta G_{DZ}$ for all of the data points except the outlier $\alpha_1$V279A given $P_{o\text{-}GABA\text{-}max} = 1$. The good description of the data suggests that DZ's energetic contribution to pore gating is the same for all of the constructs on this line, estimated as $\Delta\Delta G_{DZ} = -0.4$ kcal/mol. In contrast, $\alpha_1$V279A more than doubles the energy that DZ binding transmits to the pore gate. A comparison of the bold and light gray symbols shows that reducing $P_{o\text{-}GABA\text{-}max}$ to 0.5 primarily shifts the data points along the fitted line with only minor changes in $\Delta\Delta G_{DZ}$, indicating that our assumption for the value of $P_{o\text{-}GABA\text{-}max}$ is not critical for interpretation of $\Delta\Delta G_{DZ}$. Nonetheless, we verified that single L9'T/V279A receptors open with high probability (***Figure 7—figure supplement 1***). (C) Summary of $\Delta\Delta G_{DZ}$ for individual oocytes. Negative values of $\Delta\Delta G_{DZ}$ increase channel open probability. Gray box plots indicate the median and 25th and 75th percentiles. The vertical line is the position of linear fit in (B) corresponding to $-0.4$ kcal/mol.

The online version of this article includes the following figure supplement(s) for figure 7:

**Figure supplement 1.** Single-channel opening for L9'T/V279A in saturating GABA.

equilibrium independent of $\alpha_1$M2–M3 linker mutation with the notable exception f $\alpha_1$V279A that more than doubles $\Delta\Delta G_{DZ}$ (***Figure 7B,C*** and ***Table 1***). Importantly, this result is largely independent of our assumption for $P_{o\text{-}GABA\text{-}max}$ (see bold vs. light gray symbols in ***Figure 7B***). Nonetheless, we verified that single L9'T/V279A channels in saturating GABA open with high probability to a similar maximal conductance level as wild-type and L9'T channels (***Figure 7—figure supplement 1***). These observations suggest that coupling between the BZD site and pore gate is relatively independent of individual $\alpha_1$M2–M3 linker side chain interactions except for $\alpha_1$V279, which natively hinders DZ gating as compared to its substitution with alanine. Strikingly, introduction of a bulky tryptophan or charged aspartate at position 279 have much smaller effects on $\Delta\Delta G_{DZ}$, consistent with the idea that the small side chain volume of alanine is the most relevant factor for increased DZ efficiency.

## The mutation $\alpha_1$V279A enhances DZ potentiation of GABA-evoked currents in a wild-type background

If gating in the $\alpha_1$L9'T background is relevant to neurotransmitter-driven gating in native receptors, then the mutation $\alpha_1$V279A should both impair gating by GABA and enhance DZ potentiation of GABA-evoked currents in a wild-type background. To test this, we compared GABA-evoked current responses in $\alpha_1\beta_2\gamma_{2L}$ (wild-type) and $\alpha_1$V279A$\beta_2\gamma_{2L}$ (V279A) receptors, as well as the ability of 1 µM DZ to potentiate these responses. First, GABA $EC_{50}$ is increased by $\alpha_1$V279A, consistent with impaired unliganded gating (***Figure 8A***). This effect is similar, although slightly larger than previously observed in $\alpha_2\beta_1\gamma_{2S}$ receptors (***O'Shea and Harrison, 2000***). Second, DZ potentiates GABA-evoked peak current responses to a greater extent in V279A than in wild-type receptors, consistent with enhanced coupling between the BZD site and pore gate in the mutant (***Figure 8B,C***). In contrast to wild-type, DZ also potentiates V279A currents evoked with a saturating concentration of GABA. Together, these data suggest that the additional DZ modulation conferred by $\alpha_1$V279A is both independent of agonist association and additive to DZ potentiation in wild-type receptors.

In addition to the BZD positive modulator DZ, we examined the effect of the $\alpha_1$V279A mutation on the BZD negative modulator FG-7142 (***Im et al., 1995***). Responses to ~$EC_{20\text{-}25}$ GABA were reduced in the presence of FG-7142 in both WT and V279A receptors, but to a slightly lesser extent in V279A (***Figure 8—figure supplement 1***). Distinct mechanisms for BZD positive and negative modulation could explain the observed reduction in negative modulation as compared to enhancement of positive modulation conferred by $\alpha_1$V279A. Alternatively, our observations are also consistent with the idea that $\alpha_1$V279A stabilizes an open channel configuration specifically in the BZD-bound complex and that this stabilization works in concert with positive modulators but hinders negative modulators.

## $\alpha_1$L9'T and $\alpha_1$V279A have independent and additive effects on the pore-gating equilibrium

To further explore the idea that $\alpha_1$V279A confers its effects primarily by altering pore gating in DZ-bound receptors, we asked whether a simple Monod–Wyman–Changeux (MWC) model of receptor

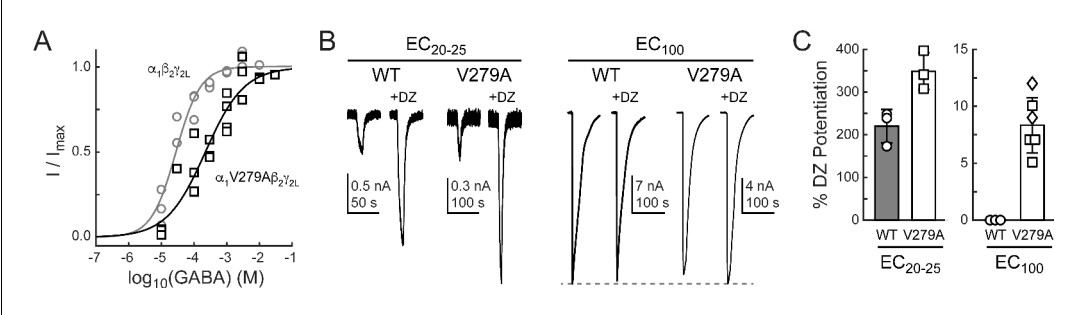

**Figure 8.** The mutation $\alpha_1$V279A enhances DZ potentiation of GABA-evoked current amplitudes in a wild-type (WT) $\alpha_1\beta_2\gamma_{2L}$ background. **(A)** Normalized GABA concentration–response relationships for WT (circles, three oocytes) and V279A (squares, four oocytes). Solid lines are fits to *Equation 1* for all oocytes combined. Fit parameters are $EC_{50}$, hill slope (# oocytes): WT = 28 μM, 1.2 (3); V279A = 222 μM, 0.7 (4). **(B)** Potentiation of current amplitudes evoked by 10 second pulses of either subsaturating $EC_{20-25}$ (WT: 10 μM, V279A: 30 μM) or saturating $EC_{100}$ (WT: 3 mM, V279A: 3–30 mM) GABA by 1 μM DZ. **(C)** Summary of potentiation as shown in (**B**) for individual oocytes. For V279A $EC_{100}$, squares indicate 3 or 10 mM GABA and diamonds 30 mM GABA.

The online version of this article includes the following figure supplement(s) for figure 8:

**Figure supplement 1.** Reduction of GABA-evoked currents by the BZD negative modulator FG-7142 for $\alpha_1\beta_2\gamma_{2L}$ (WT) and $\alpha_1$V279A$\beta_2\gamma_{2L}$ (V279A) receptors.

behavior can account for our observations (*Figure 9A*). This model has been widely used to describe pseudo-steady-state behavior in GABA$_A$ receptors (*Steinbach and Akk, 2019*; *Rüsch and Forman, 2005*; *Chang and Weiss, 1999*). For comparison with model simulations, we estimated open probability ($P_o$) as a function of GABA or DZ concentration (black curves in *Figure 9B*) from fits of *Equation 1* to the data shown in *Figures 2*, *4,* and *8* where the minimum and maximum $P_o$ were scaled according to our observed PTX-sensitive and GABA- or DZ-evoked current amplitudes. In the $\alpha_1$L9'T background, the maximum GABA-evoked $P_o$ was allowed to vary during optimization, and the minimum unliganded $P_o$ was scaled relative to the maximum by the observed mean ratio of $I_{PTX}/I_{GABA-max}$ (*Figure 3*). The maximum DZ-evoked $P_o$ was scaled relative to the minimum unliganded $P_o$ by the observed mean ratio of $I_{DZ-max}/I_{PTX}$ (*Figure 5*). The maximal $P_o$ was set to 0.85 for $\alpha_1\beta_2\gamma_{2L}$ receptors (*Keramidas and Harrison, 2010*) and allowed to vary for $\alpha_1$V279A$\beta_2\gamma_{2L}$ receptors.

The model assumes independent GABA and DZ binding. Model parameters were optimized by globally minimizing the sum of squared residuals between simulated (*Equation 4*) and estimated $P_o$ as a function of GABA or DZ concentration for $\alpha_1$L9'T$\beta_2\gamma_{2L}$, $\alpha_1$L9'T/V279A$\beta_2\gamma_{2L}$, and $\alpha_1\beta_2\gamma_{2L}$ receptors (*Figure 9B*). Responses to $\alpha_1$V279A$\beta_2\gamma_{2L}$ receptors were not considered during optimization and treated as a model prediction. The mutations $\alpha_1$L9'T and $\alpha_1$V279A were allowed to perturb the parameter L with the assumption that their effects are independent in the L9'T/V279A double mutant. For wild-type receptors, L was fixed according to its approximate relation with $EC_{50}$ as

$$L_{WT} = L_{L9T}\left(EC_{50,WT}/EC_{50,L9T}\right)^2$$ (*Steinbach and Akk, 2019*; *Akk et al., 2018*; *Scheller and Forman, 2002*; *Chang and Weiss, 1999*). The value of d was assumed to be the same for L9'T and wild-type receptors and allowed to vary in the presence of the $\alpha_1$V279A mutation. The resulting optimized model parameters (*Figure 9* caption) were highly constrained by the data (i.e. estimated $P_o$), and the model does a fairly good job of describing our observed responses to either GABA or DZ, differing somewhat in the steepness of their concentration dependence (*Figure 9B*). Furthermore, the model qualitatively accounts for observations from $\alpha_1$V279A$\beta_2\gamma_{2L}$ receptors even though they were not considered during optimization. The model's predicted maximum $P_o$ approached 1.0 in the $\alpha_1$L9'T background, consistent with a gain of function phenotype. In contrast, maximal $P_o$ was reduced in $\alpha_1$V279A$\beta_2\gamma_{2L}$ as compared to wild-type receptors, as expected for a reduction in unliganded activity. These results suggest that $\alpha_1$L9'T and $\alpha_1$V279A have independent and additive effects on the gating equilibrium, and that for $\alpha_1$V279A, this effect depends on DZ occupancy.

Although responses to GABA or DZ alone are largely explained by this model, DZ potentiation of GABA-evoked responses in a wild-type background is either underestimated or overestimated for subsaturating or saturating GABA concentrations, respectively (*Figure 9C*). Previous applications of

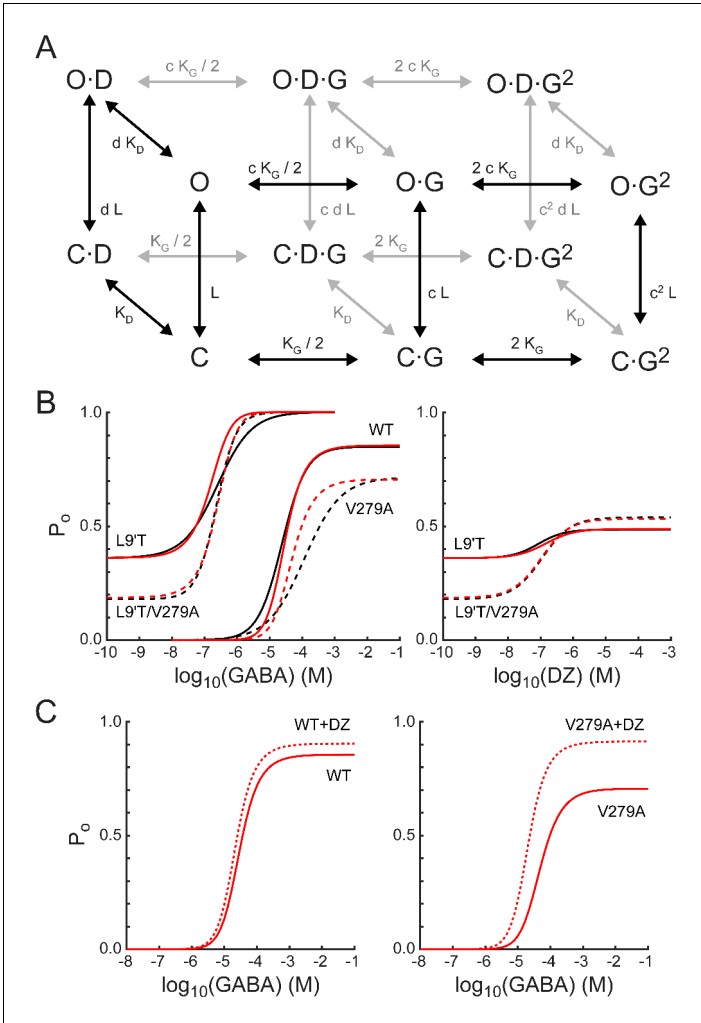

**Figure 9.** A simple MWC model of channel gating largely accounts for the observed effects of $\alpha_1$L9'T and $\alpha_1$V279A via independent and additive effects on the pore-gating equilibrium. (**A**) The model depicts channel gating between closed (C) and open (O) states with independent binding of two GABA (G) and one DZ (D) molecule. L is the ratio of closed to open state probabilities in the absence of ligand, $K_G$ and $K_D$ are the respective dissociation constants for GABA or DZ, and c and d are the respective factors by which GABA or DZ binding influence channel opening. The probability to be in an open state is given by **Equation 4**. (**B**) Estimated open probability ($P_o$) from the data in **Figures 2–5** (black) overlaid with model simulations (**Equation 4**, red). See main text for a detailed description. Model parameters are: $L_{L9'T} = 1.8$; $L_{L9'T/V279A} = 4.4$; $L_{WT} = 18000$; $K_G = 53$ μM; $K_D = 180$ nM; c = 0.0031; d = 0.59; $d_{V279A} = 0.20$. (**C**) The model's prediction for potentiation of WT and V279A GABA-evoked responses by 1μM DZ.

The online version of this article includes the following source data and figure supplement(s) for figure 9:

**Source code 1.** MWC simulation code.

**Figure supplement 1.** Estimated open probability ($P_o$) from the data in **Figure 2** (black) overlaid with simulations for the model in **Figure 9A** (red).

---

this model to BZD modulation in gain of function backgrounds predict a larger left shift of the $P_o$ curve in the presence of drug more similar to that observed, in part due to a smaller value of d (**Campo-Soria et al., 2006**; **Downing et al., 2005**; **Rüsch and Forman, 2005**). However, reducing d also enhances the predicted potentiation of responses to saturating GABA, contrary to that observed for wild-type receptors. Thus, the model is either too simplistic or its assumptions too strict to describe all aspects of DZ potentiation in wild-type channels that both rapidly desensitize and may involve drug modulation of agonist binding and/or intermediate gating steps (**Goldschen-**

*Ohm et al., 2014*; *Gielen et al., 2012*). Nonetheless, the model does predict an increase in DZ potentiation of responses evoked by both subsaturating and saturating GABA for $\alpha_1$V279A, qualitatively similar to that observed. For responses to saturating GABA in V279A, 42% of the predicted potentiation is accounted for by the lower $P_{o\text{-GABA-max}}$ as compared to WT, and the remainder reflects the mutations enhancement of DZ gating.

## Effects of alanine mutations on gating by GABA

To explore the effects of the mutations on gating by GABA, we fit estimated open probability versus GABA concentration relationships using the model in *Figure 9A* (front face only in the absence of DZ). Estimated open probabilities from fits of *Equation 1* to the data in *Figure 2* were scaled according to the assumption that $P_{o\text{-GABA-max}} = 1$ as described above. We first asked whether all of the mutations could be explained solely by differences in their intrinsic closed–open equilibrium (parameter L). *Figure 9—figure supplement 1A* shows model predictions where all constructs share identical affinities for GABA (i.e. parameters $K_G$ and c). The overall rough qualitative agreement with the data suggests that differences in unliganded open probability are likely to account for much of the observed effects. For comparison, we next constrained L based on the estimated unliganded $P_o$ as $L = (1 - P_o)/P_o$ and allowed GABA affinity to differ across constructs. The resulting model fits were slightly improved to a nearly equivalent extent regardless of whether the relative affinity for closed versus open states (parameter c) was held constant (*Figure 9—figure supplement 1B*) or allowed to vary (*Figure 9—figure supplement 1C*). Thus, the simplest conclusion is that mutations with right-shifted EC$_{50}$s reduce GABA affinity for the closed state, although we cannot rule out compensatory changes in affinity for closed and open states.

## Discussion

The main conclusions of this study are as follows: First, in the $\alpha_1$L9'T gain of function background alanine substitutions in the $\alpha_1$M2–M3 linker generally impair unliganded pore opening, indicating that side chain interactions with the linker are important for gating even in the absence of bound agonist. Second, the same mutations have no effect on the amount of chemical energy from DZ binding transmitted to the pore gate, except for $\alpha_1$V279A which more than doubles DZ's energetic contribution to pore gating. Thus, $\alpha_1$V279 plays a crucial role to natively hinder drug modulation as compared to its substitution with alanine. Third, introduction of a bulky tryptophan or charged aspartate at position 279 is less favorable than the smaller alanine, suggesting that DZ modulation is inhibited by side chain interactions at the center of the linker. Fourth, $\alpha_1$V279D severely impairs unliganded gating. Fifth, $\alpha_1$V279A similarly enhances DZ potentiation of GABA-evoked currents in a wild-type background. Sixth, the effects of $\alpha_1$V279A in both $\alpha_1$L9'T and wild-type backgrounds are explained by specific changes in the pore gating equilibrium and its coupling to DZ binding at the BZD site.

The use of gain of function mutations to study modulatory or weakly activating ligands is well appreciated (*Germann et al., 2019*; *Akk et al., 2018*; *Campo-Soria et al., 2006*; *Downing et al., 2005*; *Rüsch and Forman, 2005*; *Findlay et al., 2001*). Single-channel gating dynamics for combinations of gain of function mutations in AChR suggest that the unliganded and agonist-bound gating mechanisms are similar if not identical and also that they are similarly affected by mutations (*Purohit and Auerbach, 2009*). Consistent with this idea, MWC models of pseudo steady-state GABA$_A$R function have been largely successful in describing the effects of gain of function mutations with changes to the pore gating equilibrium independent of agonist binding (*Steinbach and Akk, 2019*; *Downing et al., 2005*; *Campo-Soria et al., 2006*; *Rüsch and Forman, 2005*; *Chang and Weiss, 1999*). The model reported here in *Figure 9* supports this conclusion.

Although the effects of $\alpha_1$V279A can also be explained by specific changes to the pore-gating equilibrium, they are dependent on DZ occupation of the BZD site. From the model in *Figure 9A*, we compute an efficiency for GABA of $\eta_{GABA} = [1 - log(K_G)/log(cK_G)] \times 100\% = 37\%$, similar to that reported previously (*Nayak et al., 2019*). In contrast, DZ efficiency is $\eta_{DZ} = [1 - log(K_D)/log(dK_D)] \times 100\% = 3\%$. Whereas $\alpha_1$V279A decreases unliganded pore opening, it also increases DZ efficiency ($\eta_{DZ,V279A} = 9\%$) to the extent that DZ binding overcomes its intrinsic inhibitory effect and results in similar or even enhanced activation as compared to wild type (*Figure 9*). Interestingly, an M2–M3 linker mutation in the $\gamma_2$ subunit was similarly observed to impair

activation by GABA and enhance modulation by the general anesthetic propofol (*O'Shea et al., 2009*).

Our estimates for the energetic effect of DZ binding on pore gating (−0.4 kcal/mol) are similar to prior estimates from direct gating in L9'S/T backgrounds and from kinetic models (*Goldschen-Ohm et al., 2014*; *Gielen et al., 2012*; *Downing et al., 2005*; *Rüsch and Forman, 2005*). This is approximately 10-fold less than the energy derived from binding each molecule of GABA (*Goldschen-Ohm et al., 2014*; *Maksay, 1994*; *Jackson, 1992*). However, it is sufficient to produce a relevant change in open probability in channels with small free energy differences between closed and open states, such as conferred by single bound agonists, partial agonists, other allosteric modulators, or gain of function mutations. From a clinical perspective, such small perturbations are likely to be more easily tolerated in patients. Given that the M2–M3 linker is associated with early movements during AChR activation (*Purohit et al., 2013*), it is possible that biasing the linker toward its active conformation in the $\alpha_1$L9'T background could limit our ability to observe its full range of motion. In this case, our observed effects of linker mutations on DZ modulation may underestimate their effects on the full activation pathway.

The interface between the extracellular agonist- and BZD-binding domains and the transmembrane helices is formed largely by several loops including the M2–M3 linker, Cys loop, β1–β2 loop and β8–β9 linker, and pre-M1 segment from neighboring subunits (*Figure 1*). Mutagenesis suggests that interactions between these domains are important for channel gating by agonists (*Kash et al., 2003*; *O'Shea and Harrison, 2000*) thought to involve an outward radial displacement of the M2–M3 linker (*Nemecz et al., 2016*). Rate versus free energy linear relationships in AChR suggest that the M2–M3 linker moves early during channel activation, similar to rearrangements at agonist binding sites (*Purohit et al., 2013*). Consistent with these observations, mutant cycle analysis indicates strong long-range coupling between residues in the M2–M3 linker and agonist binding sites (*Gupta et al., 2017*). Given the homology between the BZD site at the $\alpha/\gamma$ interface and agonist sites at $\beta/\alpha$ interfaces, interactions between the M2–M3 linker and BZD site are reasonable, although they need not be very strong given the relatively small energy DZ contributes to pore gating. As with agonist sites, such interactions could be mediated by global backbone conformational fluctuations or via distinct structural components, or both. Either way, side chain interactions at position 279 in the $\alpha_1$ subunit play an important role in coupling between the BZD site and pore gate.

In structural models, the M2–M3 linker adopts a C-shaped conformation with $\alpha_1$V279 oriented inwards toward the center of the arc (*Figure 1D*; *Kim et al., 2020*; *Laverty et al., 2019*; *Masiulis et al., 2019*; *Phulera et al., 2018*; *Zhu et al., 2018*). We hypothesize that the side chain at position 279 is centrally involved in steric interactions between linker residues near the top of the M2 and M3 helices. Removing this obstruction ($\alpha_1$V279A) would allow the linker to become more compact and bring the M2 and M3 helices closer together. In contrast, similar or larger side chains ($\alpha_1$V279D/W) would sterically inhibit such a conformational change. In this case, we speculate that the closer proximity of the M3 helix could impair unliganded pore gating by hindering radial expansion of the pore lining M2 helix. Conversely, a more compact transmembrane domain may also enhance coupling with the BZD site (*Kim et al., 2020*). Valine and threonine residues in GlyR and AchR aligning with V279 (*Figure 1—figure supplement 1*) adopt a similar conformation (*Kumar et al., 2020*; *Nemecz et al., 2016*; *Du et al., 2015*), suggesting that this residue has a conserved role in other pLGICs.

Alternatively, removing most of the central side chain may simply alter linker flexibility. Linker flexibility has been suggested to be inversely correlated with gating, where stabilizing interactions between $\alpha_1$R19' (located just below $\alpha_1$V279) and the linker backbone may promote coupling between extracellular and intracellular domains (*Masiulis et al., 2019*). Thus, the strong inhibition of unliganded opening conferred by $\alpha_1$V279D could reflect competition for $\alpha_1$R19', thereby weakening its interaction with the linker backbone and increasing linker flexibility. However, this does not provide a simple explanation for the opposing effects of $\alpha_1$V279A on unliganded versus DZ-bound gating. Regardless, it is important to keep in mind that the energetic changes that we observe for DZ gating are on the order of a hydrogen bond or two, and thus can be accounted for by relatively subtle changes.

The β8-β9 linker and pre-M1 in the neighboring $\gamma_2$ subunit come in close proximity to the $\alpha_1$M2–M3 linker and are known to be important for BZD modulation, with the β8-β9 linker also contributing to the BZD-binding site (*Hanson and Czajkowski, 2011*; *Hanson and Czajkowski, 2008*). In AChR

strong coupling with the adjacent β8-β9 linker and pre-M1 segment of the neighboring subunit occurs primarily via the threonine aligning with $\alpha_1$V279 in GABA$_A$R and its neighboring serine (*Gupta et al., 2017*). A reorientation of the linker due to removal of steric interactions near its center could alter coupling to the BZD site via interactions with the $\gamma_2$ β8-β9 linker and/or pre-M1. For example, a compression of the ends of the M2–M3 linker would cause the middle of the linker to be squeezed outward toward the neighboring $\gamma_2$ subunit, potentially enhancing intersubunit coupling. Comparison of GlyR structures in apo or antagonist-bound versus agonist-bound conformations indicates that intersubunit interactions between the M2–M3 linker in the vicinity of the valine aligning with V279 and the following serine, and the top of the M1 helix in the adjacent subunit are weakened during channel activation (*Du et al., 2015*; *Kumar et al., 2020*). Thus, a stronger intersubunit coupling in this area could potentially both reduce unliganded opening and enhance coupling with rearrangements of the $\gamma_2$ subunit upon BZD binding. However, other regions including the β4-β5 linker in the channel's outer vestibule also affect BZD modulation of agonist-evoked currents (*Pflanz et al., 2018*; *Venkatachalan and Czajkowski, 2012*). Thus, BZD modulation likely involves larger domain fluctuations in addition to any specific molecular pathways involving $\alpha_1$V279.

Recent structures of $\alpha\beta\gamma$ receptors show DZ bound not only at the classical $\alpha/\gamma$ site in the extracellular domain, but also in the transmembrane domain below the M2–M3 linker at $\beta/\alpha$ and $\gamma/\beta$ intersubunit interfaces (*Kim et al., 2020*; *Masiulis et al., 2019*). Thus, it is intriguing to speculate that enhanced DZ gating as conferred by $\alpha_1$V279A may involve occupation of a transmembrane site. However, these structures were solved in the presence of 100–200 μM DZ, and the relevance of these transmembrane sites to high affinity binding of ~1 μM DZ at the classical site is unclear (*Walters et al., 2000*). Also, DZ binding in the transmembrane domain was not observed at $\alpha/\gamma$ or $\alpha/\beta$ interfaces. Therefore, we favor the interpretation that our observed effects reflect altered functional coupling with the classical site.

Our observations reveal a critical residue V279 in the $\alpha_1$M2–M3 linker, which regulates energetic coupling between the BZD site and the pore gate, whereas other linker side chains contribute little or not at all. These data shed new light on the molecular basis for GABA$_A$R modulation by one of the most widely prescribed classes of psychotropic drugs.

## Materials and methods

### Mutagenesis and expression in oocytes

DNA for wild-type GABA$_A$R rat $\alpha_1$, $\beta_2$, and $\gamma_{2L}$ subunits were a gift from Dr. Cynthia Czajkowski. Single alanine substitutions were introduced throughout the $\alpha_1$M2–M3 linker from L276-T283 in addition to the gain of function $\alpha_1$L9'T pore mutation (rat $\alpha_1$L263T) (QuikChange II, Qiagen). Mutations V279D and V279W were introduced similarly. Each construct was verified by forward and reverse sequencing of the entire gene. mRNA for each construct was generated (mMessage mMachine T7, Ambion) for expression in *X. laevis* oocytes (EcoCyte Bioscience, Austin, TX). Oocytes were injected with 27–54 ng of total mRNA for α, β, and γ subunits in a 1:1:10 ratio (*Boileau et al., 2002*) (Nanoject, Drummond Scientific). Oocytes were incubated in ND96 (in mM: 96 NaCl, 2 KCl, 1 MgCl$_2$, 1.8 CaCl$_2$, 5 HEPES, pH 7.2) with 100 mg/ml gentamicin at 18˚C.

### Two-electrode voltage clamp recording and analysis

Currents from expressed channels 1–3 days post-injection were recorded in two-electrode voltage clamp (Dagan TEV-200, HEKA ITC and Patchmaster software). Oocytes were held at −80 mV and perfused continuously with buffer (ND96) or buffer containing PTX, GABA, or DZ. PTX was diluted from a 1 M stock solution in DMSO. DZ was diluted from a 10 mM stock solution in DMSO. Fresh PTX and DZ stock solutions were tested several times with no change in results. GABA was dissolved directly from powder. A microfluidic pump (Elveflow OB1 MK3+) and rotary valve (Elveflow MUX Distributor) provided consistent and repeatable perfusion and solution exchange across experiments, which limited solution exchange variability to primarily differences between oocytes only. Ten second pulses of PTX, GABA, or DZ were followed by 3–6 min in buffer to allow currents to return to baseline. Recorded currents were analyzed with custom scripts in MATLAB (Mathworks). Recordings of concentration–response relationships were bookended by pulses of PTX to correct for any drift or rundown during the experiment. Briefly, currents were baseline subtracted to correct for

drift and then scaled by a linear fit to the peak of each PTX response to correct for rundown (*Figure 2—figure supplement 1*). The amount of current rundown was variable across oocytes, with no clear relation to specific constructs. Concentration–response relationships were fit to the Hill equation:

$$\frac{I}{I_{max}} = \frac{1}{1 + \left(\frac{EC_{50}}{[X]}\right)^n}$$

(1)

where $I$ is the peak current response, $[X]$ is ligand concentration, $EC_{50}$ is the concentration eliciting a half-maximal response, and $n$ is the Hill slope.

## Single-channel recording

HEK293T cells were transfected with DNA for rat $\alpha_1$L9'T/V279A, $\beta_2$ and $\gamma_{2L}$ subunits in a 1:1:1 ratio. Single-channel currents were recorded 16–32 hr post-transfection from excised inside-out patches clamped at −80 mV. Currents were acquired at 20 kHz and low-pass filtered at 2 kHz (Axopatch 200A, HEKA ITC, and PatchMaster software). Extracellular (pipet) solution was (in mM): 145 NaCl, 2.5 KCl, 2 CaCl$_2$, 1 MgCl$_2$, 10 HEPES, 4 Dextrose, pH 7.3 with NaOH. Intracellular (bath) solution was (in mM): 140 KCl, 10 EGTA, 10 HEPES, 2 MgATP, pH 7.3 with KOH. GABA was dissolved in the extracellular solution.

## DZ-gating model

For the model in *Figure 7A*, the free energy difference for unliganded ($\Delta G_0$) and DZ-bound ($\Delta G_1$) gating was estimated as follows:

$$\Delta G_0 = -kT \ln\left(\frac{P_{open}}{P_{closed}}\right)_{unliganded} \approx -kT \ln\left(\frac{P_{o-GABA-max}(I_{PTX}/I_{GABA-max})}{1 - P_{o-GABA-max}(I_{PTX}/I_{GABA-max})}\right)$$

(2)

$$\Delta G_1 = -kT \ln\left(\frac{P_{open}}{P_{closed}}\right)_{DZ-bound} \approx -kT \ln\left(\frac{P_{o-GABA-max}(I_{DZ-max}/I_{GABA-max})}{1 - P_{o-GABA-max}(I_{DZ-max}/I_{GABA-max})}\right)$$

(3)

where $I_{PTX}$, $I_{GABA-max}$, and $I_{DZ-max}$ are as shown in *Figures 3* and *5*, $k$ is the Boltzmann constant, $T$ is temperature, and $P_{o-GABA-max}$ is the open probability in saturating GABA, which was assumed to be approximately 1. Importantly, even if this assumption is incorrect, the effect on DZ's energetic contribution to pore gating should be minimal and our overall conclusion unchanged (see *Figure 7B*). Also, this assumption was verified for single L9'T/V279A receptors (*Figure 7—figure supplement 1*).

## MWC model

For the model in *Figure 9A*, the probability to be in an open (O) state is given by:

$$P_o = \left(1 + L\frac{\left(1 + \frac{[DZ]}{K_D}\right)\left(1 + \frac{[GABA]}{K_G}\right)^2}{\left(1 + \frac{[DZ]}{dK_D}\right)\left(1 + \frac{[GABA]}{cK_G}\right)^2}\right)^{-1}$$

(4)

where L is the ratio of closed (C) to open (O) state probabilities in the absence of ligand, $K_G$ and $K_D$ are the respective dissociation constants for GABA or DZ, and c and d are the respective factors by which GABA or DZ binding influence pore opening.

## Acknowledgements

We thank Dr. Cynthia Czajkowski for gifting DNA for wild-type GABA$_A$R subunits, and Drs. Richard Aldrich and Eric Senning for helpful discussions and shared use of equipment.

## Additional information

### Funding

| Funder | Grant reference number | Author |
|---|---|---|
| University of Texas at Austin | Department of Neuroscience Startup | Marcel P Goldschen-Ohm |
| University of Texas at Austin | STARS | Marcel P Goldschen-Ohm |

The funders had no role in study design, data collection and interpretation, or the decision to submit the work for publication.

### Author contributions

Joseph W Nors, Data curation, Software, Formal analysis, Validation, Investigation, Visualization, Writing - original draft, Writing - review and editing; Shipra Gupta, Resources, Investigation; Marcel P Goldschen-Ohm, Conceptualization, Data curation, Software, Formal analysis, Supervision, Visualization, Methodology, Project administration, Writing - review and editing

### Author ORCIDs

Marcel P Goldschen-Ohm (iD) https://orcid.org/0000-0003-1466-9808

### Decision letter and Author response

Decision letter https://doi.org/10.7554/eLife.64400.sa1
Author response https://doi.org/10.7554/eLife.64400.sa2

## Additional files

### Supplementary files

• Transparent reporting form

### Data availability

Source code for the modeling depicted in Figure 9 has been provided.

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
