## [Decision Letter]

**Acceptance summary:**

In this work, the authors make use of a gain of function mutant in the GABA-A receptor to study the mechanism of action of benzodiazepines (BDZ), not actual activators, but allosteric modulators of the function of the receptor. This work is important because the action of allosteric modulators is often difficult to study in the absence of a full ligand, which makes studying allosteric coupling extremely difficult. Since BDZ have important clinical applications, this work should further our understanding of the mechanisms of these important drugs. The work identifies a critical amino acid residue which regulates the actions of BDZs. This is an important finding, which opens new avenues to understand the molecular biophysics of GABA receptors.

**Decision letter after peer review:**

Thank you for submitting your article "A critical residue in the α_1_M2-M3 linker regulating GABA_A_ receptor pore gating by diazepam" for consideration by *eLife*. Your article has been reviewed by three peer reviewers, including Leon D Islas as the Reviewing Editor and Reviewer #1, and the evaluation has been overseen by Kenton Swartz as the Senior Editor. The following individual involved in review of your submission has agreed to reveal their identity: Ryan E Hibbs (Reviewer #3).

The reviewers have discussed the reviews with one another and the Reviewing Editor has drafted this decision to help you prepare a revised submission.

Summary:

Nors et al. have investigated the effects of mutations to the M2-M3 linker in the α1 subunit on α1β2γ2 GABA-A receptor gating by GABA and diazepam. For simplicity, the experiments were conducted on the L9'T background that enhances unliganded gating and enables measurement of current in the presence of diazepam alone. Replacement of each individual residue (6 in total) by alanine indicated that the V279A enhances gating efficacy by diazepam while other substitutions were largely without effect. Comparison of alanine, tryptophan and aspartate substitutions at V279 suggested that side chain volume at this position is critical to gating by diazepam. Overall, the experiments are expertly conducted and support major conclusions. The manuscript is very well organized and written, but it raises questions that could be easily addressed in the confounds of this study.

A major conclusion of the present manuscript suggests that another class of ligands binding in the intersubunit interface in the extracellular domain employs the M2-M3 linker in energy transduction. This finding is interesting but not unexpected. Furthermore, the finding is not particularly well studied. However, the study includes important advances that will be of interest to the LGIC community, but could be improved through consideration of additional experiments, and more careful pruning of hypotheses presented in the Discussion after looking at GlyR (or other) structures.

Essential revisions:

1) Why were not all M2-M3 linker residues mutated? Specifically, the 280 and 282 positions, but beyond these the M2-M3 linker extends as far as F288. Also, please provide a sequence alignment for the M2-M3 linker region for α1, β2, and γ2 subunits.

2) The authors should test the effect of V279A on BZD negative modulators. Is inverse agonism similarly affected?

3) Testing a few key positions in the γ subunit M2-M3 linker would make the story stronger. Many questions are left unanswered – does the γ subunit M2-M3 linker affect gating by diazepam? Does it affect gating by GABA? In any case, the effects of the Ala substitutions in the α subunit should be formally analyzed for gating by GABA. These data are already available, just no analysis was done.

4) If V279 is so important for gating and energy transduction, its interacting counterpart should be defined. Molecular modeling should help determine the nearby residues in ECD, then mutate V279 and the counterparts individually and in combination, followed by double mutant cycle analysis.

5) A main assumption in this work is that all mutants will have the same or very similar open probability to the background channels and to WT channels gated by GABA. This assumption might seem reasonable but should be checked experimentally through single channel recording, especially for the 279A channels.

6) It is important to fully justify the two corrections applied to the data that are shown in Figure 2—figure supplement 1. Do all channels show a constant reduction of current size during the experiment? If not, is there evidence of gating mode-shifting during the recording period? What is the reason that in the second linear scaling, the size of the last current is chosen as the normalizing factor?

7) Subsection “α_1_L9’T and α_1_V279A have independent and additive effects on the pore gating equilibrium”: the model assumes independent GABA and DZP binding. Is this reasonable? Their binding is cooperative, isn't it, with DZP binding shifting the affinity for GABA a few fold to the left (higher affinity)?

8) Setting up the first section in the Results with a clear rationale for looking at unliganded gating would improve readability.

9) Some direct experimental comparison of WT and L9'T, including extent of ptx-sensitive leak, would be helpful.

10) Discussion: it may be worthwhile to look at structures of the related glycine receptor, which have been obtained in resting, open and closed conformational states- is the V279 “obstruction” doing something interesting that you can make more broadly relevant, as well as place it in the context of experimental structures? Do the mutagenesis results (Trp, Asp mutants) make sense in the context of state changes in these comparable structures? Or are the GlyR structures not useful for refining hypotheses on this topic?

---

## [Author Response]

Essential revisions:1) Why were not all M2-M3 linker residues mutated? Specifically, the 280 and 282 positions, but beyond these the M2-M3 linker extends as far as F288. Also, please provide a sequence alignment for the M2-M3 linker region for α1, β2, and γ2 subunits.

Positions 280 and 282 are both alanine natively in α1, and thus were not mutated for the alanine scan. Regarding the extension of the M2-M3 linker as far as F288, we simply restricted our analysis to the flexible loop region between the helical regions associated with M2 and M3 as resolved in recent structures. We have amended the text to make these points clear. We have also added Figure 1—figure supplement 1 which shows a sequence alignment of the M2-M3 linker regions as requested.

2) The authors should test the effect of V279A on BZD negative modulators. Is inverse agonism similarly affected?

We have added a new supplementary figure, Figure 8—figure supplement 1, showing the effects of the BZD negative modulator FG-7142 on WT and V279A receptors. We have also added a paragraph in the Results describing these data.

3) Testing a few key positions in the γ subunit M2-M3 linker would make the story stronger. Many questions are left unanswered – does the γ subunit M2-M3 linker affect gating by diazepam? Does it affect gating by GABA? In any case, the effects of the Ala substitutions in the α subunit should be formally analyzed for gating by GABA. These data are already available, just no analysis was done.

We agree with the reviewers that looking at the effect of M2-M3 linker mutations in other subunits such as g would be interesting. However, to do this well would require at least another entire scan equivalent to that currently reported. Even with a key residue such as V279, there are multiple combinations with a, b and g subunits that should be examined. We feel that this would be more appropriate as its own follow-up investigation.

Regarding effects of the mutations on gating by GABA, we have appended an additional Results section and new supplementary figure, Figure 9—figure supplement 1 describing application of the model in Figure 9A to GABA P_o_ curves for all alanine mutations.

4) If V279 is so important for gating and energy transduction, its interacting counterpart should be defined. Molecular modeling should help determine the nearby residues in ECD, then mutate V279 and the counterparts individually and in combination, followed by double mutant cycle analysis.

We appreciate the desire to identify specific interaction counterparts. However, double mutant cycle analysis using current responses can lead to inaccurate estimates of the interaction energies if the closed and open reference states are not the same for all mutants (see PMC4210424), which is a distinct possibility in this case. Furthermore, the energies we are talking about here are fairly small, with V279A’s effect on DZ gating being only -0.4 kcal/mol. The clear effect on channel current of such small energetic perturbations is made possible by the small energy difference between closed and open states in the L9'T mutant. Resolving the details of weak interactions on this scale that may also involve interactions with backbone is likely to be nontrivial and we feel would constitute a study in its own right.

5) A main assumption in this work is that all mutants will have the same or very similar open probability to the background channels and to WT channels gated by GABA. This assumption might seem reasonable but should be checked experimentally through single channel recording, especially for the 279A channels.

We agree with the reviewers. We have added a new supplementary figure, Figure 7—figure supplement 1 that shows data for a single L9'T/V279A channel in saturating GABA. Po during prolonged bursting is ~0.93, consistent with the assumed high open probability of constructs in the L9'T background. We have also added a Materials and methods section for these recordings. Due to the experimental challenge in obtaining these data, we restricted this verification to V279A only.

6) It is important to fully justify the two corrections applied to the data that are shown in Figure 2—figure supplement 1. Do all channels show a constant reduction of current size during the experiment? If not, is there evidence of gating mode-shifting during the recording period? What is the reason that in the second linear scaling, the size of the last current is chosen as the normalizing factor?

The amount of current rundown observed was variable across oocytes, with no clear relation to specific constructs. There was no clear evidence of a gating mode shift that we could discern from our recordings. Also, responses for the same construct were similar after correction for oocytes with either clearly obvious or nearly non-existent rundown. We have added a statement to this effect in the Materials and methods. For the second linear scaling, the choice of the last current as the normalizing factor is arbitrary as all currents were normalized to their PTX response for comparison across oocytes.

7) Subsection “α_1_L9’T and α_1_V279A have independent and additive effects on the pore gating equilibrium”: the model assumes independent GABA and DZP binding. Is this reasonable? Their binding is cooperative, isn't it, with DZP binding shifting the affinity for GABA a few fold to the left (higher affinity)?

For the MWC model in Figure 9A, cooperativity stems from the different affinities in closed versus open states. Thus, even though binding steps are independent, DZ binding will alter the equilibrium between closed and open states and thereby alter overall GABA binding, and vice versa. Nonetheless, as we acknowledge in the text, this model does not perfectly describe observations for WT channels with DZ and GABA in combination, and thus a more complicated mechanism may be required. Regardless, the simple model used here does do a reasonable job of describing our observations in the L9'T background.

8) Setting up the first section in the Results with a clear rationale for looking at unliganded gating would improve readability.

We agree that this will be helpful for readers and have added some explanation to the beginning of the first Results section.

9) Some direct experimental comparison of WT and L9'T, including extent of ptx-sensitive leak, would be helpful.

We have added a new supplementary figure, Figure 2—figure supplement 3 which illustrates WT responses to GABA, PTX and DZ.

10) Discussion: it may be worthwhile to look at structures of the related glycine receptor, which have been obtained in resting, open and closed conformational states- is the V279 “obstruction” doing something interesting that you can make more broadly relevant, as well as place it in the context of experimental structures? Do the mutagenesis results (Trp, Asp mutants) make sense in the context of state changes in these comparable structures? Or are the GlyR structures not useful for refining hypotheses on this topic?

The GlyR structures are certainly of value for refining hypotheses regarding pLGICs. As such, we now discuss them briefly in the Discussion. The GlyR structures in general paint a very similar picture to that from GABA_A_R and AChR structures. Thus, the role of position 279 is likely conserved in other pLGICs, and we have added a sentence aimed at broadening the relevance for this site (Discussion). In our opinion, the structures do not however make the exact mechanism for each of the mutations completely clear. Given the relatively small energetic effects that we observe and the resolution of these models, it is likely the case that additional experiments or MD simulations will be needed to resolve these details.